# Gold Smuggling in India and Its Effect on the Bullion Industry

**Maria Immanuvel Susai** [1,*] and **Lazar Daniel** [2]

1   St. Joseph's Institute of Management, 28/1 Primrose Road, Off M G Road, Bangalore 560025, India
2   Department of Commerce, School of Management, Pondicherry University, Puducherry 605014, India;
    lazar.dani@gmail.com
*   Correspondence: mariaimmanuvel@sjim.edu.in; Tel.: +91-94876-29563

**Abstract:** This study strives to examine when and where most of the gold smuggling takes place in India. It further analyses the causal relationship between smuggled gold and other macroeconomic variables. Finally, it analyses how the smuggled gold affects the Indian bullion industry. The data related to gold smuggling has been sourced from the website of the Directorate Revenue Intelligence and analysed using graphs and the Granger causality test. The variables used in the study are the quantity of smuggled gold, exchange rates, the major stock indices in the world, the number of auspicious days in a month, domestic and international gold prices, India's jewellery export, the GDP, customs duty, and the domestic gold supply. The results revealed that most of the gold smuggling takes place on Fridays and mostly occurs in the months of October, November, and December. The states of West Bengal, Delhi, Maharashtra, and Tamil Nadu account for most of the gold smuggling in India. A positive correlation is observed between the smuggled gold, India's gold demand, the number of auspicious days in the month, India's jewellery export, India's GDP, India's domestic gold supply, and stock indices such as SENSEX, FTSE100, DFMGI. Gold smuggling in India is caused by India's gold demand, the level of jewellery export, the GDP, domestic and international gold prices, and India's customs duty.

**Keywords:** gold smuggling; exchange rates; economy; stock indices; jewellery export; customs duty; gold price

**JEL Classification:** G1; E7E2

## 1. Introduction

India is the second-largest consumer of gold in the global gold market. Gold imports are treated as legitimate when the shipments arrive with a certificate of mining origin. Gold entering the country without a proper record of origin and not through authorized agencies and the 22 nominated banks is called "grey" importing (Labh 2015). An estimate says that around 700 kg of gold enters India illegally every year (Ray 2014). Smugglers are using innovative ways to bring in the metal illegally. Gold is melted into seed-shaped chips and hidden in dates or capsules, or ground into granules and mixed with other metals to look like ore 22 (Hardikar 2014). Sometimes, gold is converted into gold belt buckles and torch batteries and concealed in bags, clothes, and smugglers' rectums (Madhukalya 2021; Goudreau 2019; Indianexpress 2020a, 2020b, 2020c; Ray 2014). The Directorate of Revenue Intelligence (DRI) and Enforcement Directorate officers have found passengers hiding gold in ingenious places, including in toys, chewing-gum packets, sewing machines, wheelchair frames, and suitcase liners (Indianexpress 2021; Pisharody 2020; Goudreau 2019). Grey-market operators and businesses are able to sell gold at a more discounted price than the authorized agents in the domestic markets as they avoid paying the customs duty (Kallungal 2018).

Figure 1 clearly depicts the comparison between the gold that comes in the formal importing process and the illegal flow of gold. It is evident from the graph that the

smuggled gold follows the same pattern as the official gold import. When the official gold import increased, the volume of smuggled gold also increased.

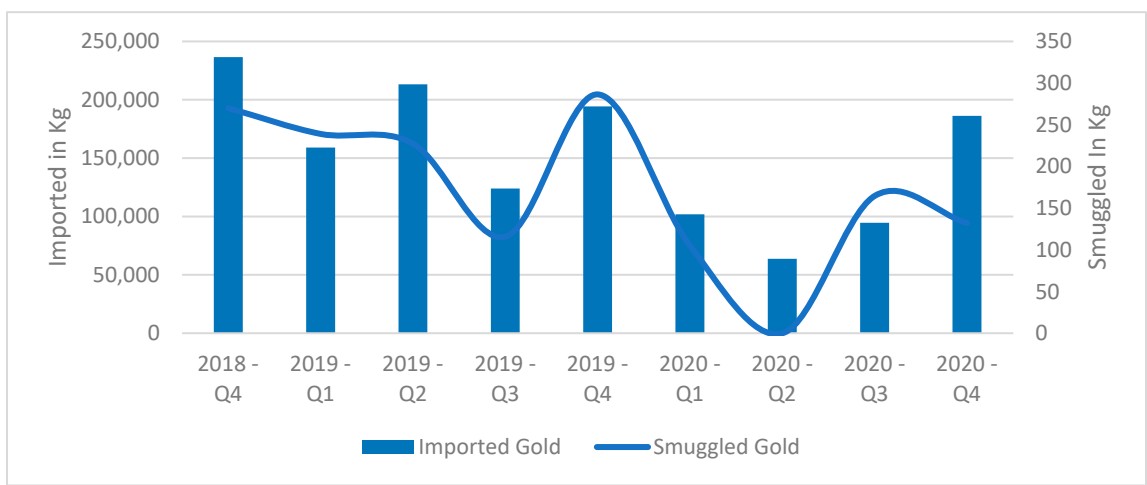

**Figure 1.** Comparison between imported gold and smuggled gold. Source: compiled and prepared by the researchers from https://www.gold.org/, https://dri.nic.in/ accessed on 30 March 2022.

It is estimated that up to one-fourth of the total volume of gold entering India arrives through illicit trade. India imports around 800–900 tonnes of gold every year, while the annual consumption is around 1000 tonnes. This suggests that up to 200 tonnes of gold is being smuggled into the country annually (Dutta 2020). This illicit trade represents over USD 1 billion in value and at least USD 20 million in lost tax revenue to governments (Jayakumar 2020; Dasari 2018). The World Gold Council (WGC) estimates that 65–75% of smuggled gold comes by air, 20–25% by sea, and 5–10% by land (World Gold Council Report 2017).

One important factor that encourages the smuggler is the customs duty levied on gold imports (Bloomberg 2019, 2021; Bullionvault 2017). History shows that there has been a considerable increase in the percentage of customs duty. PR Somasundaram, Managing Director of the World Gold Council—India, said that the propensity to smuggle now is very high because every time the tax rate is increased, much greater incentive is given to smugglers (Bloomberg 2019; Livemint 2019; Roy 2019; Jadhav 2016). As per the present market value of gold in India, 1 kg of the smuggled yellow metal would fetch a profit of more than INR 5 lakh on import duty alone (Philip 2020; Bundhun 2019).

Data shared by the Ministry of Finance showed that in the last three fiscal years, the maximum amount of smuggled gold was seized from international airports in Chennai, Kozhikode, Kochi, Mumbai, and Delhi. Further, it shows that out of the top 10 airports from where the maximum quantity of smuggled gold was seized, the top three international airports are in Kerala (Mallapur 2020). Dubai is still the number one city from where gold is smuggled, and Singapore is slowly emerging as another one (Deol 2019; Nair 2020; Reuters 2019a). Sri Lanka has also become a staging point (Ananthalakshmi and Mayenkar 2013). Sometimes, airline staff also extend their help to gold smugglers coming from Gulf countries (Indianexpress 2021). With one third of the world's gold passing through its borders (Sikarwar 2018), India has established itself not only as one of the leading gold manufacturing centres but also as one of the world's largest smuggling hubs (Martin 2019). UAE is its primary source of smuggled refined bullion (Martin 2019; News18 2019).

Table 1 and Figures 2 and 3 provide detailed information about gold smuggling in India over the last 6 years. It is observed from the data that the quantity of gold seized kept increasing over that period of time. Furthermore, the number of cases of smuggling at various airports and the number of people involved in gold smuggling also kept increasing. During the years 2020–2021, these were very minimal due to the COVID-19 restrictions imposed in the country. International travellers were much fewer due to limited flight operations and other COVID-19 restrictions implemented during this period.

**Table 1.** Gold smuggling in India.

| Year | Number of Cases of Smuggling of Gold at Various Airports | Quantum of Gold Seized (in Kg) | Number of People Booked | Value of Gold Seized (Lakhs in INR) |
|---|---|---|---|---|
| 2015–16 | 2696 | 2452.147 | 1408 | 60,667.29 |
| 2016–17 | 1453 | 921.805 | 788 | 24,375.62 |
| 2017–18 | 2911 | 1996.930 | 1525 | 53,133.32 |
| 2018–19 | 4855 | 2946.097 | 2141 | 83,354.89 |
| 2019–20 | 4444 | 2629.549 | 2339 | 85,795.50 |
| 2020–21 * | 196 | 103.165 | 200 | 4955.566 |

\* up until August 2020. Source: Government of India, Ministry of Finance, Department of Revenue, Lok Sabha unstarred Question No. 28.

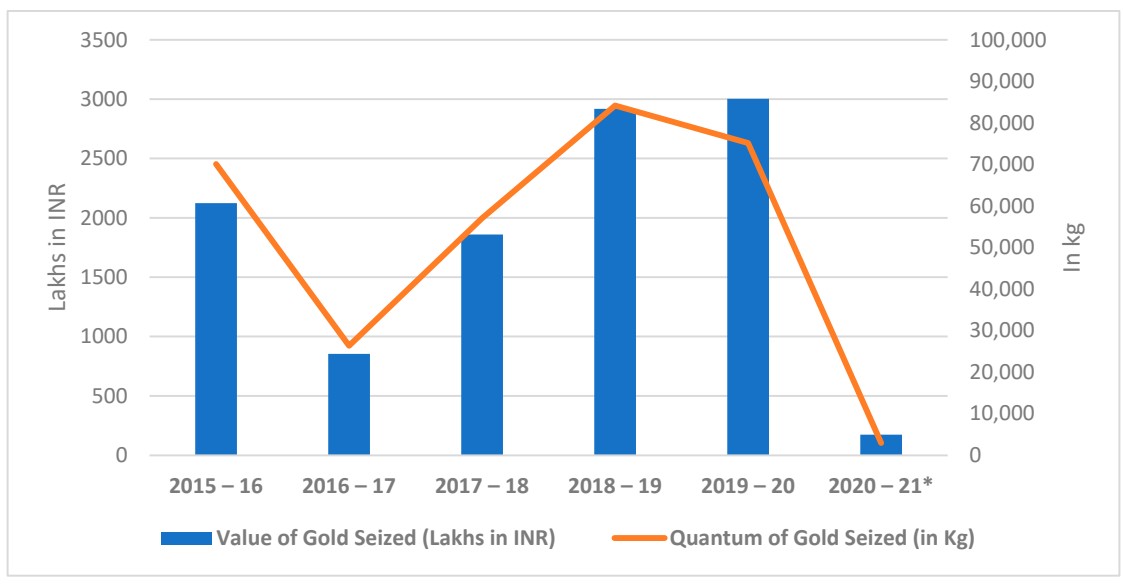

**Figure 2.** Quantum and value of gold seized. * up until August 2020. Source: Government of India, Ministry of Finance, Department of Revenue, Lok Sabha unstarred Question No. 28, compiled and prepared by the researchers.

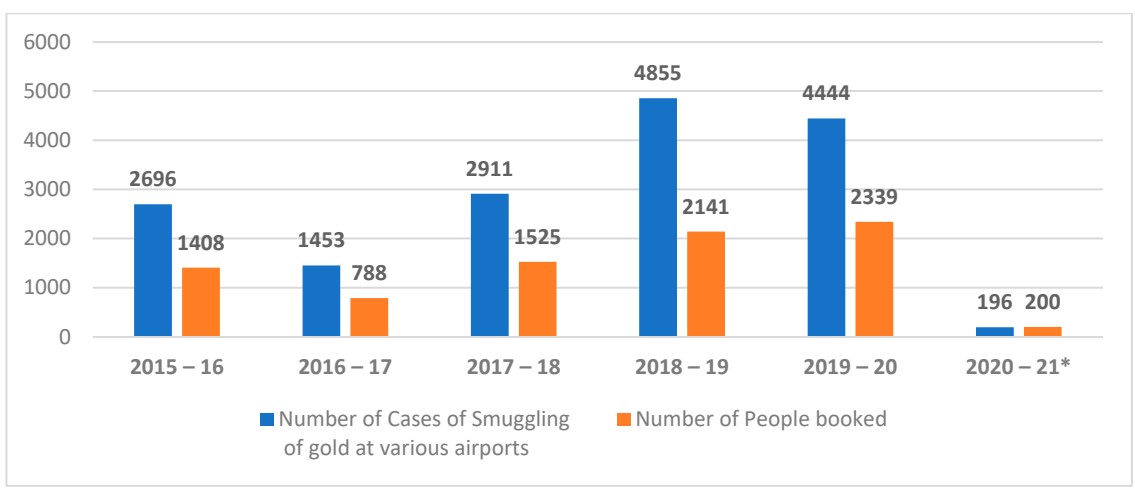

**Figure 3.** Number of cases and people booked for gold smuggling. * up until August 2020. Source: Government of India, Ministry of Finance, Department of Revenue, Lok Sabha unstarred Question No. 28, compiled and prepared by the researchers.

## 2. Literature Research

Almost one third of India's annual gold demand is fulfilled through illegal gold. Nevertheless, it is hard to find research in this field. There are a few studies that have documented the opinions of industry experts about gold smuggling in India. Mahadevan (2020) has exhaustively covered the illicit gold flow to India. The report discussed in detail about the history of illicit gold flow into the country and the various methods and routes used by the smugglers. The study concluded that high customs duties and the abundance of black money encouraged smugglers to bring gold illegally into the country (Bundhun 2019; Roy 2019). Furthermore, the lack of attractive investment opportunities makes people buy illegal gold at a cheaper rate.

Another important research study was undertaken by Martin (2019) from Impact, an independent non-profit organization. This study focused on how India became one of the world's largest gold smuggling hubs. It examined the link between India and the gold trade in South America and Africa. It extensively studied how the illicit gold produced in South American and African countries flowed into India. In addition to this, it discussed the policy vulnerabilities and drivers that made India become a hub for illicit gold. The key findings of the study are that traders create fake documents for importing gold ore illegally from Africa and South America into India, and that may be linked to conflict, human rights abuses, illegality, and criminal networks. Refined bullion is smuggled into India from other trading centres, notably the UAE (Jadhav 2016; Bureau 2019; Deol 2019). The trade of illicit gold—both from Uganda to the UAE and from the UAE to India—is often financed through hawala. Illicit gold enters the country, is absorbed into the legal market with ease, and is re-exported back as jewellery. The limitation of the above study is the methodology used in the research report. The author interviewed experts in the industry and documented their opinions on gold smuggling in the report. No data were collected and no statistical analysis was made to verify the veracity of the information.

Buehn and Farzanegan (2012) undertook a study on smuggling around the world. Using a multiple indicator, multiple cause (MIMIC) model to analyse the determinants of smuggling, their study revealed that high levels of corruption and a low rule of law encourages smuggling. Tariffs and trade restrictions are important push factors, while a higher black market premium (BMP) discourages smugglers. Based on the MIMIC analysis, their study calculated an index of smuggling which provided a ranking for 54 countries. It further concluded that smuggling was rampant in Cameroon, Pakistan, and Kenya, while it was least prevalent in Switzerland, Finland, and Sweden. A report published by the Thought Arbitrage Research Institute (TARI) (2016) focused on the top five products smuggled into India; it was found that the smuggling of gold in India was primarily driven by the demand and supply gap. The Indian gold-jewellery industry is almost completely dependent on imported raw materials, and about 90% of the requirements are fulfilled by imports. The large domestic market provides smugglers enough arbitrage (Kumar 2017) to fulfil the needs of the market through smuggled goods as the domestic gold supply is limited.

From the review of the above literature, it was found that studies have focused on the illicit flow of gold from only a few specified countries, as well as analysed how India has become the largest hub for smuggled gold, but no studies have been found on the impact of gold smuggling on the bullion market. Such studies are not based on any empirical evidence; rather, all of them are based on experts' opinions from the bullion industry. No empirical analysis was applied to support their opinions. Industry experts and government are of the opinion that it is the level of customs duty that encourages gold smuggling into the country. There may be other variables having a strong link with gold smuggling. Hence, a detailed study was undertaken to identify factors other than customs duty that might have a relationship with gold smuggling. This study is an attempt to identify and analyse the trend in gold smuggling with respect to various economic variables; using this trend, the study tries to examine when and where gold smuggling takes place in the country. In addition to this, it analyses the correlation and causal relationships between gold smuggling and other variables included in the study.

### 3. Data, Variables, and Methodology

*3.1. Methodology for the Research Work*

The study has used secondary data collected from various sources for the period from September 2018 to December 2020 for the analysis. The variables were classified into different categories, and the volume and value of gold smuggling data were compared and analysed with all the variables individually. The variables used in the analysis are the smuggled volume of gold (in kilograms), the smuggled gold value (in INR), the value of India's gold import, GDP, and jewellery export, the number of auspicious days in a month, India's customs duty, the number of national holidays in a month, etc. Gold price variables are the Mumbai price, international prices, and the price spread between domestic and international prices (Kumar 2017). Gold demand variables are the jewellery demand, bar and coin demand, and the consumer gold demand in India, the Middle East, the USA, and Hong Kong. The USA, the Middle East, and Hong Kong have been included as they are India's major jewellery export destinations. The major stock market indices considered in the analysis are the -Sensex (India), Nikkei 225 (Japan), FTSE100 (London), S&P 500 and NASDAQ (USA), CAC 40 (France), and DFMGI (Dubai). Gold supply variables included are India's scrap gold supply, India's other gold supply., Finally, the exchange rates of Indian rupees against major currencies like the USD, Pound, Euro, and Yen are included in the analysis.

Gold-smuggling data was sourced from the Directorate of Revenue Intelligence (DRI) website. Domestic and international gold prices were collected from the websites of Reserve Bank of India (RBI) and London Bullion Market Association (LBMA). Gold demand and supply data are extracted from the World Gold Council (WGC) website. The study extensively utilised graphical analyses of the data. In addition to this, statistical analyses like descriptive statistics were used to describe the nature of the data. Further, correlation was used to analyse the relationship between the variables. Lastly, the Granger causality test was used to study the causal relationship between the variables.

*3.2. Objectives of the Study*

The overall objective of the study is to analyse the trend in gold smuggling and how this smuggled gold is affecting its bullion industry. The specific objectives are as follows:

To assess the causal relationship between the volume of gold smuggled with other variables such as the exchange rates of the Indian rupee against the major currencies, major stock market indices, domestic and international gold prices, domestic and international gold demand and supply, change in customs duty, and major festivals in India (Mayenkar and Raj 2009; Afonso and Srivastava 2019). Finally, the outcome of the study will enhance policy measures to control the grey market operation in India.

*3.3. Granger Causality Test*

The causality test seeks to answer the simple question of the type "do changes in X cause changes in Y" (Granger 1969). If X causes Y, lags in X should be significant in the equation for Y. If this is the case and not vice versa, it would be said that X Granger-causes Y or that there exists a unidirectional causality from X to Y. On the other hand, if Y causes X, then lags in Y should be significant in the equation of X. If both sets of lags are significant, it would be said that there exists a bi-directional causality. A time series X is said to Granger-cause Y if it can be shown, usually through a series of F tests on lagged values of X, that those X values provide statistically significant information about future values of Y. Consider the series $Y_t$ and $X_t$:

$$Y_t = \sum_{j=1}^{p} A_{11,j} Y_{t-j} + \sum_{j=1}^{p} A_{12,j} X_{t-j} + \varepsilon_{1t} \tag{1}$$

$$X_t = \sum_{j=1}^{p} A_{21,j} Y_{t-j} + \sum_{j=1}^{p} A_{22,j} X_{t-j} + \varepsilon_{2t} \tag{2}$$

where p is the maximum number of lagged observations included in the model. The matrix A contains the coefficient of the model, and $\varepsilon_1$ and $\varepsilon_2$ are residuals for each time series. If

the variance of $\varepsilon_1$ is reduced by the inclusion of the X terms in the first equation, then it is said that X Granger-causes Y. In other words, X Granger-causes Y if the coefficient in $A_{12}$ are jointly significantly different from zero. This can be tested by performing an F-test of the null hypothesis that $A_{12} = 0$, given assumptions of covariance stationarity on Y and X. The magnitude of a Granger causality interaction can be estimated by the logarithm of the corresponding F-statistic.

## 4. Results and Discussion

The descriptive statistics of the variables are given in Table 2. The average number of gold smuggling cases booked at various airports in the last six years was 2759, and the average number of people booked under gold smuggling cases was 1400. The quantity of gold seized ranged from 103 kg to 2946 kg. The lowest record was during the COVID-19 period.

**Table 2.** Descriptive statistics of the dependent variables included in the study.

|  | Number of Cases of Smuggling of Gold at Various Airports | Number of People Booked | Value of Gold Seized (Lakhs in INR) | Quantum of Gold Seized (in kg) |
|---|---|---|---|---|
| Mean | 2759.167 | 1400.167 | 52,047.03 | 1841.616 |
| Standard Error | 719.586 | 329.6986 | 13,137.66 | 451.1412 |
| Median | 2803.5 | 1466.5 | 56,900.31 | 2224.539 |
| Standard Deviation | 1762.618 | 807.5933 | 32,180.56 | 1105.066 |
| Sample Variance | 3,106,824 | 652,207 | 1,035,588,629 | 1,221,170 |
| Kurtosis | −0.92506 | −0.83013 | −1.17998 | −0.68165 |
| Skewness | −0.27808 | −0.41128 | −0.50277 | −0.88131 |
| Range | 4659 | 2139 | 80,839.93 | 2842.932 |
| Minimum | 196 | 200 | 4955.566 | 103.165 |
| Maximum | 4855 | 2339 | 85,795.5 | 2946.097 |
| Sum | 16,555 | 8401 | 312,282.2 | 11,049.69 |

Figure 4 shows the gold smuggling trend on weekdays. It clearly indicates that most of the gold smuggling activity takes place on Fridays. On Fridays, more than double the amount of gold is smuggled than on other days in a week. On all other days, the quantity of gold smuggled is almost similar, except for Mondays.

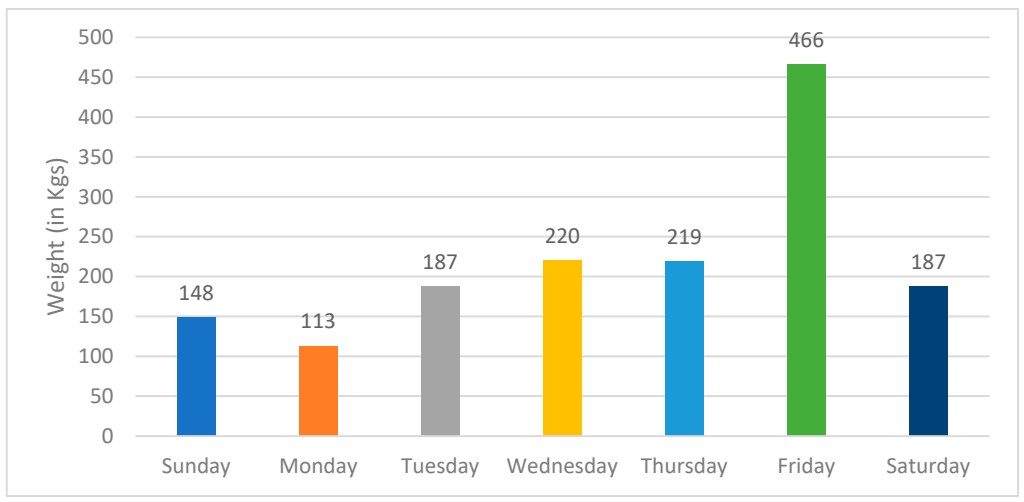

**Figure 4.** Gold smuggling trend for weekdays. Source: https://www.dri.nic.in/, compiled and prepared by the researchers (accessed on 30 March 2022).

Figure 5 supports the above conclusion as, in the last three years, all Fridays registered a higher quantity of gold than other days in the week. Also, the quantity of gold smuggled was comparatively higher in the year 2019 than in the other years. Smugglers travel to India mostly on Fridays.

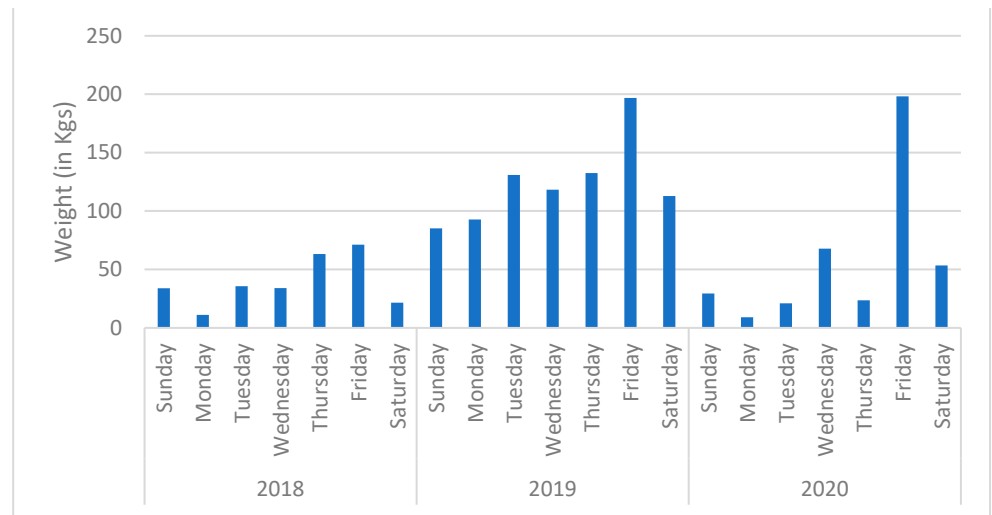

**Figure 5.** Gold smuggling trend for weekdays, arranged by year. Source: https://www.dri.nic.in/, compiled and prepared by the researchers (accessed on 30 March 2022).

The monthly gold smuggling trend is illustrated in Figures 6 and 7. Overall, it is very clear from the figure that most of the gold smuggling takes place in the months of October, November, and December. The least smuggling takes place in the month of July and August. These results correlate with the number of auspicious days in these months. We find many auspicious days in these months, and many festivals like Diwali, Navarathri, Christmas, etc. also fall within these months (Mayenkar and Raj 2009; Afonso and Srivastava 2019).

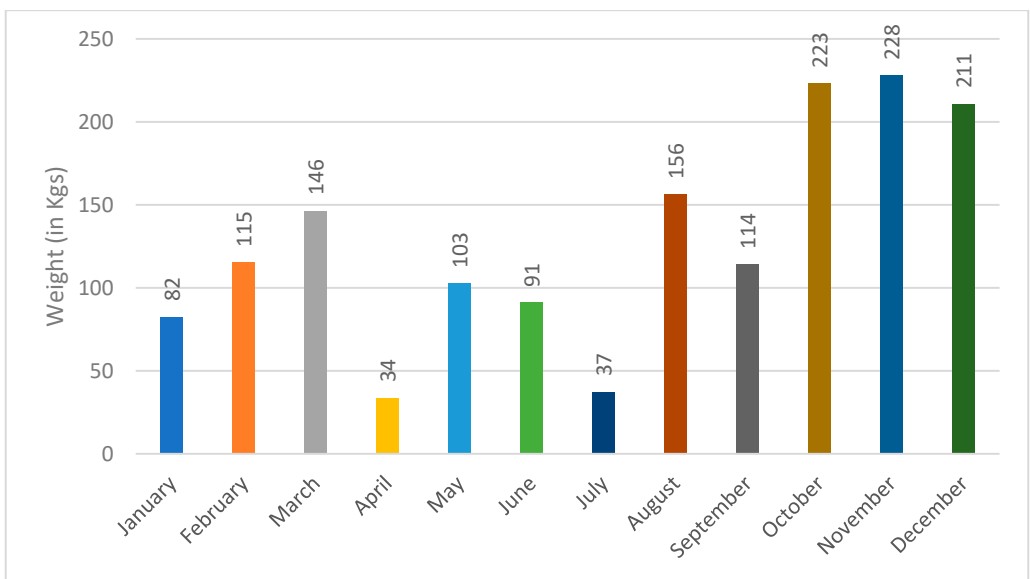

**Figure 6.** Gold smuggling trend by month. Source: https://www.dri.nic.in/, compiled and prepared by the researchers (accessed on 30 March 2022).

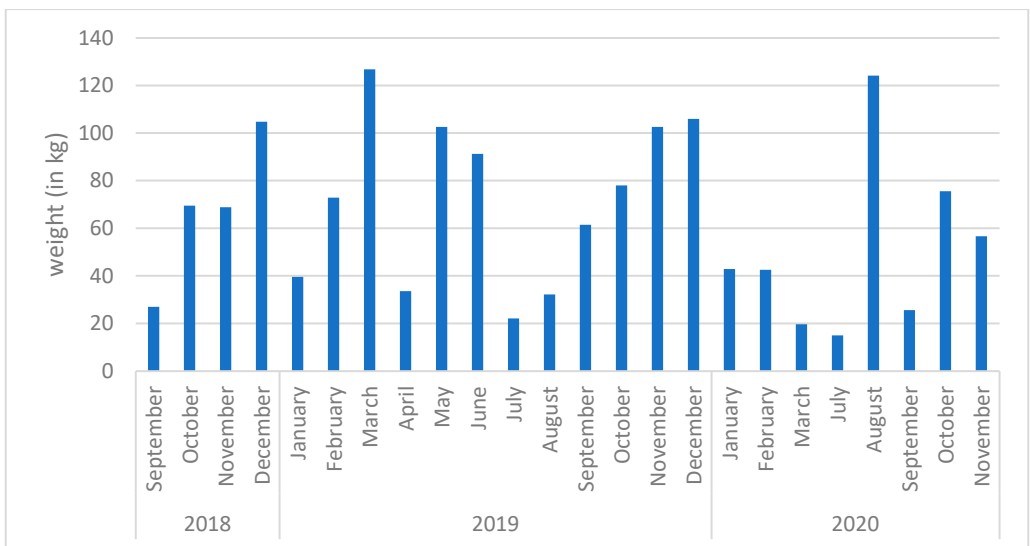

**Figure 7.** Monthly gold smuggling trend, arranged by year. Source: https://www.dri.nic.in/, compiled and prepared by the researchers (accessed on 30 March 2022).

The regions in India that account for most of the gold smuggling are outlined in Figures 8 and 9. The southern and eastern regions in India account for most of the smuggled gold, followed by the northern region. The unknown category represents missing data as the location of the gold smuggled was not found in the dataset.

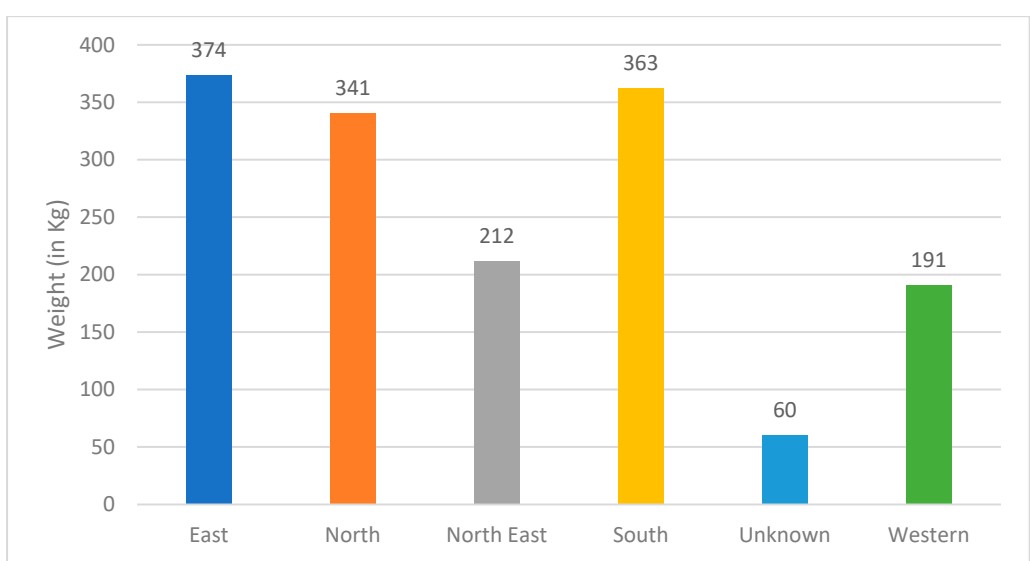

**Figure 8.** Gold smuggling trend by region. Source: https://www.dri.nic.in/, compiled and prepared by the researchers (accessed on 30 March 2022).

In the southern region, Tamil Nadu accounts for most of the smuggled gold, while in the eastern region, it is West Bengal. Figure 10 is the evidence for this conclusion. Tamil Nadu, West Bengal, Delhi, and Maharashtra are the four states in the country through which most of the gold smugglers enter the country.

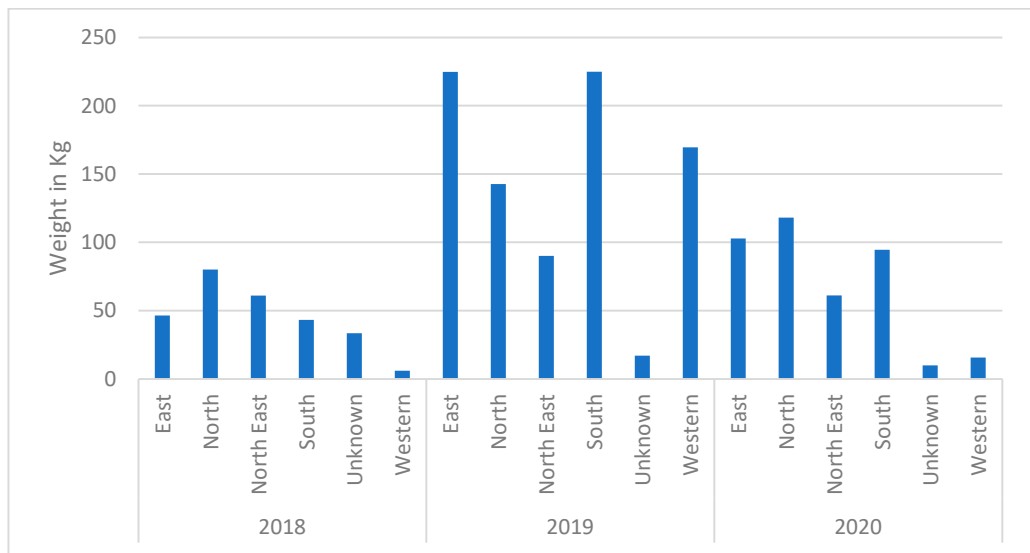

**Figure 9.** Gold smuggling trend by region and year. Source: https://www.dri.nic.in/, compiled and prepared by the researchers (accessed on 30 March 2022).

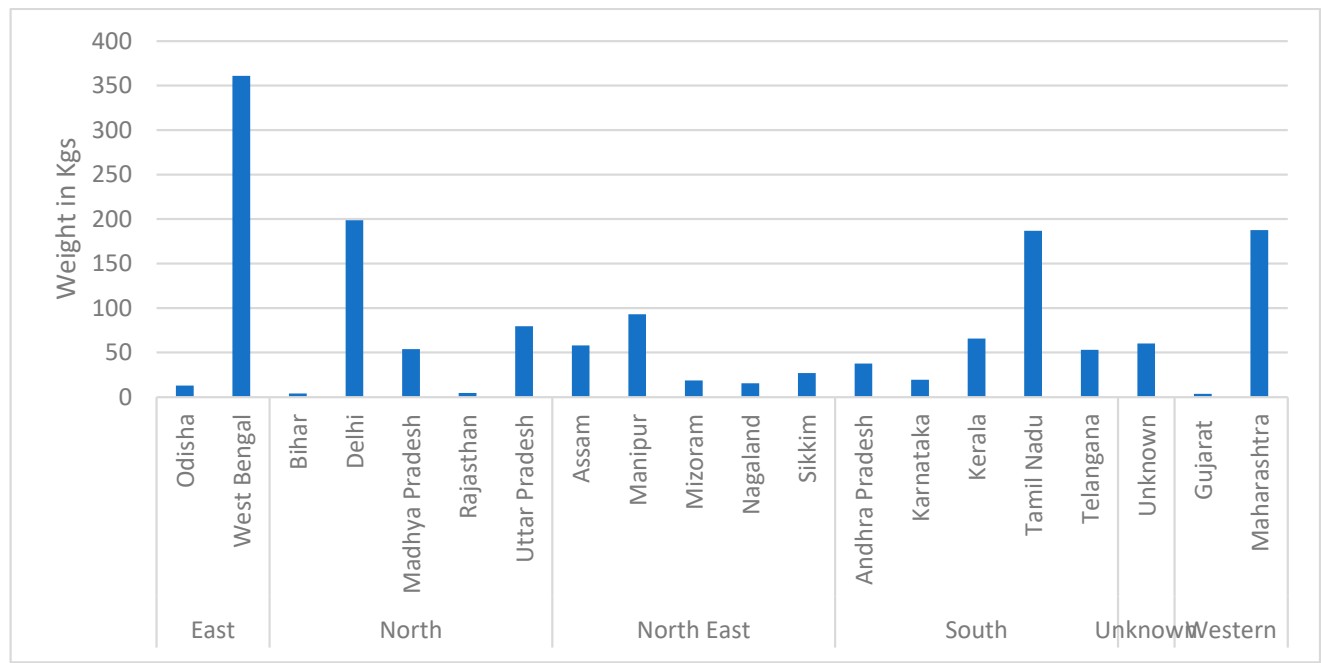

**Figure 10.** Gold smuggling trend by state. Source: https://www.dri.nic.in/, compiled and prepared by the researchers (accessed on 30 March 2022).

Figure 11 compares the value of the gold smuggled and the estimated tax loss for the government. The amount was significantly higher in 2019 than in other years (Chawla and Kesavan 2013).

Figure 12 compares the amount of smuggled gold with the number of auspicious days and national holidays in a month. It is evident from the figure that they closely follow the same pattern. The quantity of gold smuggled is higher when we have many auspicious days and national holidays in a month (Global Bullion Suppliers 2019). It is a signal to the customs authority that a greater number of auspicious days and national holidays are likely to have more smugglers bringing gold illegally into the country.

Figure 13 compares the volume of smuggled gold with fluctuations in the domestic and international gold price. It is observed that the volume of gold smuggled increases

when the domestic and international gold price decreases and vice versa. This shows that the smugglers are watching price movements closely and react accordingly to the price fluctuations to make more money. They bring the gold illegally into the country and make money by selling it on the local market when the price goes up. They take the advantage of arbitrage pricing (Kumar 2017).

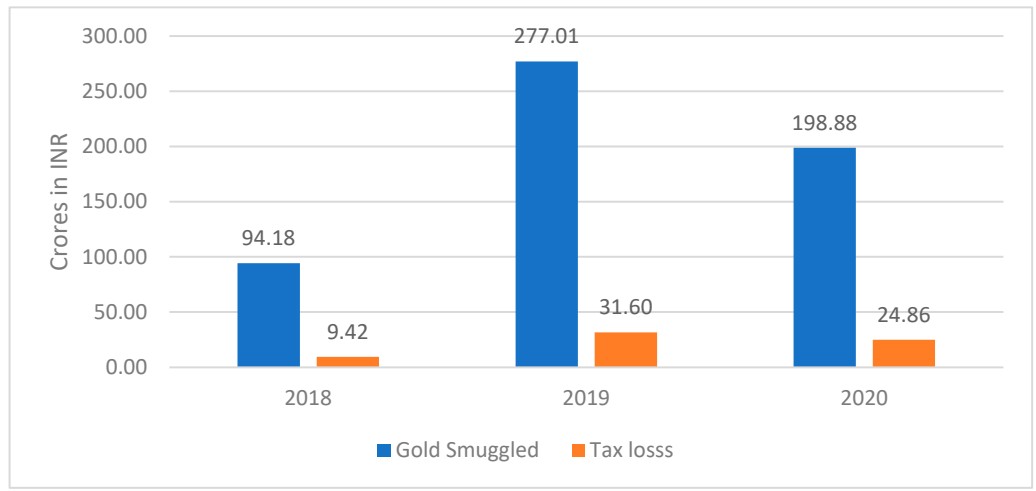

**Figure 11.** Smuggled gold value and tax loss. Source: https://www.dri.nic.in/, compiled and prepared by the researchers (accessed on 30 March 2022).

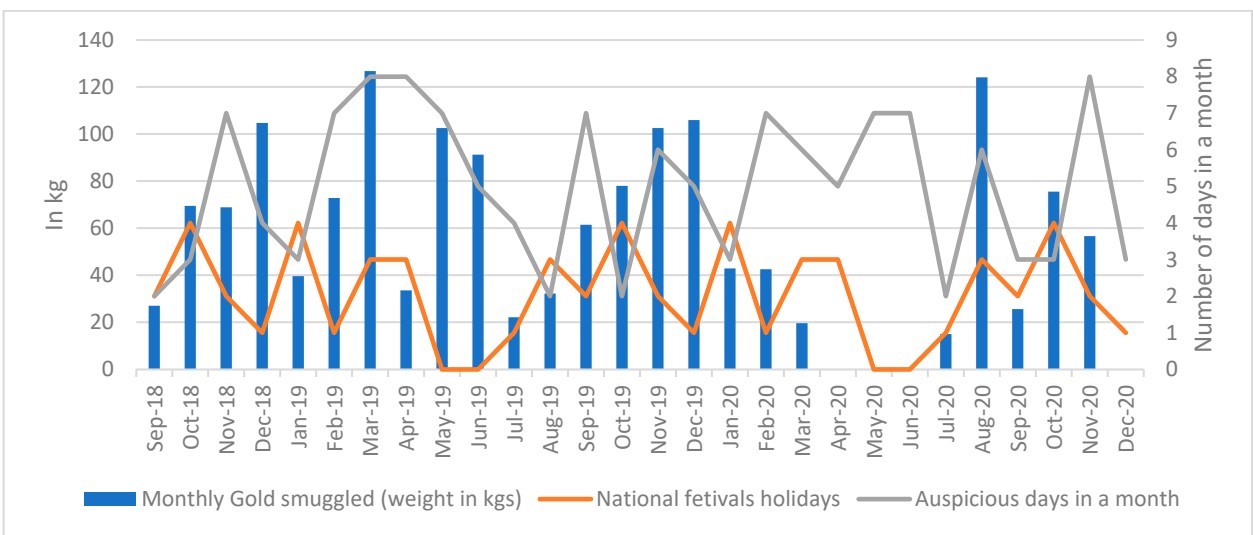

**Figure 12.** Comparison of the quantity of smuggled gold with the number of auspicious days and national holidays in a month. Source: https://www.dri.nic.in/, compiled and prepared by the researchers (accessed on 30 March 2022).

The above result is confirmed with the estimated correlation analysis. The estimated correlation between the smuggled gold and the domestic and international gold price is given in Table 3. There exists a negative weak correlation between the volume of gold smuggled and domestic and international gold prices. On the other hand, a strong positive correlation exists between the domestic and international gold price.

Figure 14 provides information about India's major jewellery export destinations. It is observed from the figure that around 90% of India's jewellery is exported to three major destinations in the world such as the USA, the Middle East, and Hong Kong. Hence this study attempted to find out how the amount of India's smuggled gold correlates with the gold demand in these destinations.

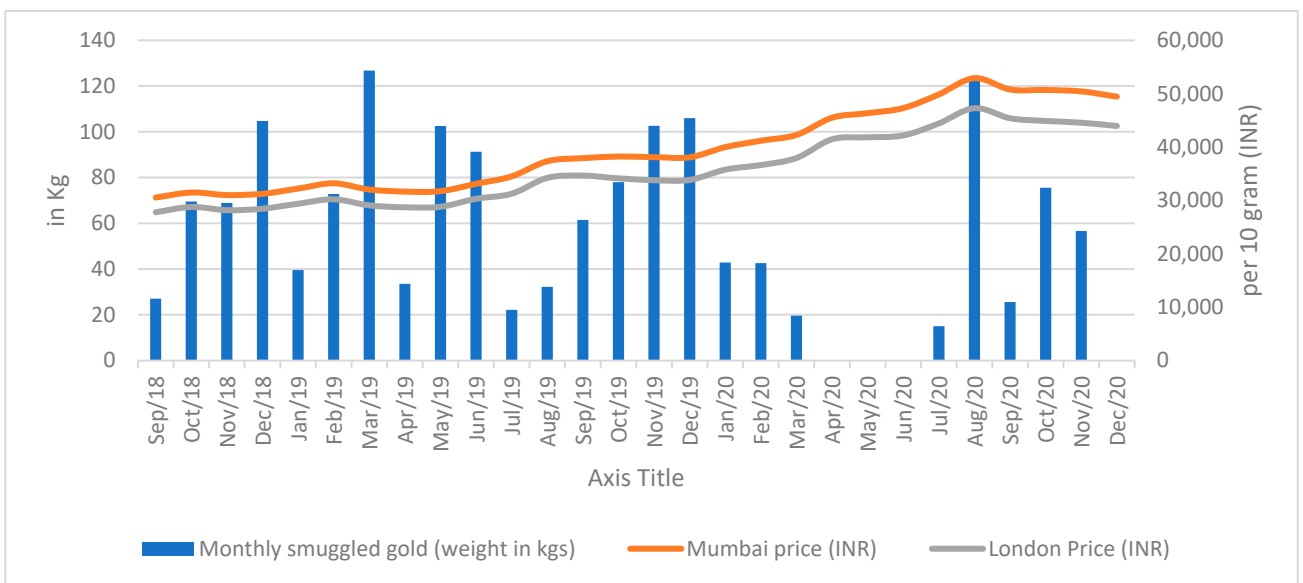

**Figure 13.** Comparison of the volume of smuggled gold with domestic and international gold prices. Source: https://www.dri.nic.in/, https://www.gold.org/, https://www.rbi.org.in/, compiled and prepared by the researchers (accessed on 30 March 2022).

**Table 3.** Correlation between smuggled gold and domestic and international gold prices.

|  | Monthly Gold Smuggled (Weight in kg) | Mumbai Price | International Price | Spread |
|---|---|---|---|---|
| Monthly gold smuggled (weight in kg) | 1 |  |  |  |
| Domestic price | −0.337 | 1 |  |  |
| International price | −0.351 | 0.999 | 1 |  |
| Spread | −0.237 | 0.961 | 0.948 | 1 |

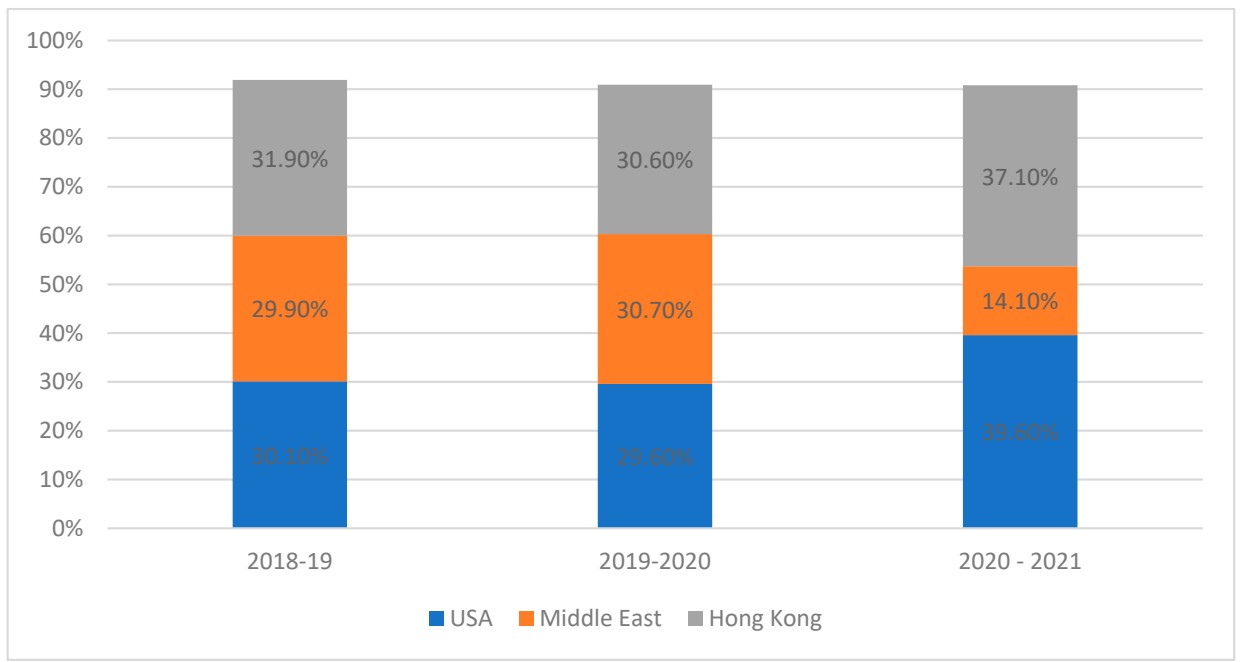

**Figure 14.** India's gem and jewellery exports. Source: https://gjepc.org/statistics.php, compiled and prepared by the researchers (accessed on 30 March 2022).

The results of the estimated correlation between India's major jewellery export destinations and the quantity of smuggled gold are given in Table 4. It shows a strong positive correlation between the quantity of smuggled gold, jewellery, and consumer demand in all the export destinations. It is inferred from the results that the gold that enters illegally into India is re-exported as gems and jewels to these major destinations (Martin 2019). Thereby, the smugglers make more money by taking advantage of arbitrage pricing (Kumar 2017).

**Table 4.** Correlation between the volume of smuggled gold and India's major jewellery export destinations.

| **INDIA** | **Smuggled Gold** |
| --- | --- |
| Smuggled Gold | 1 |
| India–Jewellery | 0.810751 |
| India–Bars and Coins | 0.7488 |
| India–Consumer | 0.81776 |
| **USA** | **Smuggled Gold** |
| Smuggled Gold | 1 |
| USA—Jewellery | 0.58298 |
| USA—Bars and Coins | −0.51157 |
| USA—Consumer | 0.28065 |
| **MIDDLE EAST** | **Smuggled Gold** |
| Smuggled Gold | 1 |
| ME—Jewellery | 0.785238 |
| ME—Bars and Coins | 0.628985 |
| ME—Consumer | 0.814034 |
| **HONG KONG** | **Smuggled Gold** |
| Smuggled Gold | 1 |
| HK—Jewellery | 0.817117 |
| HK—Bars and Coins | −0.21982 |
| HK—Consumer | 0.833414 |

The results of the correlation analysis between the volume of smuggled gold and India's jewellery export is given in Table 5. A positive correlation exists between the amount of smuggled gold and India's jewellery export. Hence, we infer that the gold that illegally enters the country positively affects its economy.

**Table 5.** Results of the correlation between smuggled gold and India's jewellery export.

| | **Monthly Gold Smuggled (Weight in kg)** | **Gold Smuggled Value (cr)** | **Jewellery Export Data** |
| --- | --- | --- | --- |
| Monthly Gold Smuggled (weight in kg) | 1 | | |
| Gold Smuggled Value (cr) | 0.823357185 | 1 | |
| India's Jewellery Export | 0.427271256 | 0.254883 | 1 |

The estimated correlation between smuggled gold and various forms of domestic gold supply is given in Table 6. It shows a strong positive correlation between the smuggled gold and the domestic gold supply. The illegal gold not only increases jewellery exports but also increases the internal gold supply in the domestic market.

**Table 6.** Results of the correlation between smuggled gold and various modes of gold supply in India.

|  | Smuggled Gold | Scrap Supply | Other Sources of Supply | India's Total Supply |
|---|---|---|---|---|
| Smuggled Gold | 1 |  |  |  |
| Scrap Supply | 0.206998 | 1 |  |  |
| Other Sources of Supply | 0.736627 | 0.366147 | 1 |  |
| India's Total Supply | 0.764728 | 0.338821 | 0.497299 | 1 |

The correlation between smuggled gold and the various components of GDP variables is given in Table 7. Except for "Financial, Real Estate, and Professional Services", all the GDP components maintain a positive relationship with the volume of gold smuggled. The overall results are also positive. This indicates that the illegal flow of gold into the country is positively affecting its economy. On the one hand, the illegal gold produces a tax loss for the government. On the other hand, it is observed that it promotes the country's jewellery exports and domestic gold supply.

**Table 7.** Results of the correlation between smuggled gold and GDP variables.

|  | Smuggled Gold |
|---|---|
| Smuggled Gold | 1 |
| Agriculture, Forestry, and Fishing | 0.353682883 |
| Mining and Quarrying | 0.323680179 |
| Manufacturing | 0.579550294 |
| Electricity, Gas, Water Supply, and Other Utilities | 0.21897382 |
| Construction | 0.648016049 |
| Trade, Hotels, Transport, Communication, and Services Related to Broadcasting | 0.620959828 |
| Financial, Real Estate, and Professional Services | −0.522026518 |
| Public Administration, Defence, and Other Services | 0.431459935 |
| Total Gross Value Added | 0.575677754 |

The volume of smuggled gold is compared with the exchange rate of the Indian Rupee against the four major currencies in the world. The correlation results are given in Table 8. The smuggled gold quantity maintains a negative relationship with the exchange rate of Indian rupees against all the major currencies. We infer from the results that the quantity of smuggled gold reduces when the Indian rupee depreciates, as it becomes expensive for smugglers. When the currency appreciates, the smugglers buy gold from the international markets and bring it illegally into the country to take advantage of arbitrage pricing.

The quantity of smuggled gold is further compared with the performance of major stock markets in the world. The correlations between the volume of smuggled gold and the major stock market indices in the world are given in Table 9. The volume of smuggled gold maintains a positive relationship with the stock market in India, UK, France, and Dubai and shows a negative relationship with the stock market in the USA and Japan (Deol 2019).

**Table 8.** Results of the correlation between smuggled gold and the exchange rate of major currencies.

|  | Monthly Gold Smuggled (Weight in kg) | Gold Smuggled Value (Crores in INR) |
|---|---|---|
| Monthly Gold Smuggled (weight in kg) | 1 | |
| Gold Smuggled Value ( Crores in INR) | 0.823357 | 1 |
| US Dollar | −0.47632 | −0.18199 |
| Pound Sterling | −0.14308 | 0.073473 |
| Euro | −0.28248 | −0.00458 |
| Japanese Yen | −0.47517 | −0.14253 |

**Table 9.** Results of the correlation between smuggled gold and major stock indices.

|  | Monthly Gold Smuggled (Weight in kg) |
|---|---|
| Monthly gold smuggled (weight in kg) | 1 |
| SENSEX | 0.253354 |
| Nikkei 225 | −0.05873 |
| FTSE100 | 0.357147 |
| S&P 500 | −0.14055 |
| CAC 40 | 0.281322 |
| DFMGI | 0.385787 |
| NASDAQ | −0.25592 |

Table 10 provides the results of the analysis of causal relationships between the variables included in the study. The pairwise Granger causality is estimated using Equations (1) and (2) given in Table 10. It shows that a bidirectional causal relationship exists between variables like the volume of smuggled gold, India's jewellery export, the domestic gold price, India's GDP, and India's jewellery demand. Unidirectional causal relationship is observed between variables like smuggled gold (Ninan 2017), India's gold import, India's consumer demand, international gold price, price spread between domestic and international gold price, stock markets of the USA, the UK and Dubai, exchange rates of Indian Rupees against USD and Yen.

The significant point to be noted here is that the customs duty affects gold smuggling. The smuggled gold Granger-causes the GDP and other important variables like India's jewellery demand and jewellery export as well as its consumer gold demand. We conclude from the results that the illegal flow of gold into the country positively affects India's bullion industry.

The study further examined the relationship between important variables like the number of auspicious days, customs duty, and domestic and international gold prices. Table 11 reveals that the number of auspicious days in a month and India's custom duty significantly Granger-cause the domestic and international gold prices and the price spread. A unidirectional causal relationship is observed between the auspicious days, India's custom duty, and both the domestic and international gold price (Bloomberg 2019, 2021; Bullionvault 2017).

**Table 10.** Results of the pairwise Granger causality test: significance of smuggled gold variable to other variables included in the study.

| Pairwise Granger Causality Test: Hypothesis | F-Statistic | Prob. |
|---|---|---|
| India's jewellery export does not Granger-cause smuggled gold | 6.13083 | 0.0207 ** |
| India's gold import does not Granger-cause smuggled gold | 13.6118 | 0.0684 *** |
| India's GDP does not Granger-cause smuggled gold | 5.10911 | 0.0733 *** |
| India's jewellery demand does not Granger-cause smuggled gold | 18.6647 | 0.0509 ** |
| Domestic price does not Granger-cause smuggled gold | 2.35487 | 0.1041 *** |
| International price does not Granger-cause smuggled gold | 2.81174 | 0.1066 *** |
| The FTSE100 does not Granger-cause smuggled gold | 2.70699 | 0.1029 *** |
| The S&P 500 does not Granger-cause smuggled gold | 3.02685 | 0.0538 *** |
| The DFMGI does not Granger-cause smuggled gold | 3.91994 | 0.0593 *** |
| The YEN does not Granger-cause smuggled gold | 3.48369 | 0.0742 *** |
| The USD does not Granger-cause smuggled gold | 2.99582 | 0.0530 *** |
| Customs duty does not Granger-cause smuggled gold | 2.48965 | 0.0908 *** |
| Smuggled gold does not Granger-cause the GDP | 8.25073 | 0.0349 ** |
| Smuggled gold does not Granger-cause price spread | 3.31755 | 0.0810 *** |
| Smuggled gold does not Granger-cause the domestic price | 9.45537 | 0.0052 * |
| Smuggled gold does not Granger-cause India's jewellery demand | 471.523 | 0.0021 * |
| Smuggled gold does not Granger-cause India's consumer gold demand | 13.4089 | 0.0694 *** |
| Smuggled gold does not Granger-cause India's jewellery export | 2.79594 | 0.0838 *** |

Significance level: * 1%, ** 5% and *** 10%.

**Table 11.** Results of pairwise Granger causality tests: significance of the number of auspicious days and customs duty to other variables.

| Pairwise Granger Causality Test: Hypothesis | F-Statistic | Prob. |
|---|---|---|
| The number of auspicious days does not Granger-cause the international gold price | 4.38175 | 0.0257 ** |
| The number of auspicious days does not Granger-cause the domestic gold price | 5.58010 | 0.0114 * |
| Customs duty does not Granger-cause the international gold price | 3.19352 | 0.0866 *** |
| Customs duty does not Granger-cause the domestic gold price | 3.63113 | 0.0688 *** |
| Customs duty does not Granger-cause the price spread between domestic and international gold prices | 2.77158 | 0.0855 *** |

Significance level: * 1%, ** 5%, and *** 10%.

Finally, the factors that Granger-cause India's jewellery export were examined and the results are given in Table 12. It was found from the results that, in addition to the smuggled gold, the performance of all the major stock markets in the world affects India's jewellery export. In addition to this, the exchange rate of the Indian rupee against the US dollar also affects India's jewellery exports.

**Table 12.** Results of pairwise Granger causality tests: significance of stock market indices and exchange rates to India's jewellery export.

| Pairwise Granger Causality Test: Hypothesis | F-Statistic | Prob. |
|---|---|---|
| The DFMGI does not Granger-cause India's jewellery export | 14.0248 | 0.0001 * |
| The FTSE100 does not Granger-cause India's jewellery export | 8.31730 | 0.0022 * |
| The S_P_500 does not Granger-cause India's jewellery export | 4.31336 | 0.0270 ** |
| The SENSEX does not Granger-cause India's jewellery export | 5.02102 | 0.0165 * |
| The CAC_40 does not Granger-cause India's jewellery export | 3.66436 | 0.0320 ** |
| The USD does not Granger-cause India's jewellery export | 3.92365 | 0.0357 *** |

Significance level: * 1%, ** 5%, and *** 10%.

## 5. Conclusions

The study aimed to analyse the pattern of gold smuggling in India and its causal relationship with other variables like the gold demand and supply in India, India's jewellery export, the exchange rate of the Indian rupee against major currencies, the major stock markets in the world, and the number of auspicious days and national holidays in a month. It was found from the results that most of the gold smuggling takes place on Fridays within the week and in the months of October, November, and December within the year. A regional analysis showed that the states of West Bengal, Delhi, Maharashtra, and Tamil Nadu account for most of the gold smuggling in India.

Results of the correlation analysis reveal that a positive relationship can be observed between smuggled gold, imported gold, and the number of auspicious days in the month (Mayenkar and Raj 2009; Afonso and Srivastava 2019). Smuggled gold shows a negative relationship with domestic and international gold prices as well as with exchange rates against the major currencies. Smuggled gold maintains a high and positive relationship with the gold demand of countries like the USA, Hong Kong, and nations in the Middle East, where most of India's jewellery export takes place.

Due to the smuggled gold, there is a tax loss for the government. However, it shows a positive correlation with India's domestic gold supply, the GDP, and India's jewellery export. Further, smuggled gold shows a positive relationship with stock indices in India (Sensex), France (FTSE100), and Dubai (DFMGI).

It was found from the pairwise Granger causality test that gold smuggling in India is caused by India's gold import, India's jewellery export, the GDP, domestic and international gold price, and the exchange rate of the Indian rupee against the USD and the Japanese Yen. India's customs duty on gold import encourages smuggling activity in the long run (Reuters 2019b), (CapitalVia 2017) (Buehn and Farzanegan 2012). Its customs duty on gold import affects its exchange rate against the major currencies and domestic and international prices. The smuggled gold affects India's GDP, its jewellery demand, and its domestic price as well. The number of auspicious days in a month in India significantly affects domestic and international gold prices. Finally, all the stock indices, the exchange rate of the Indian rupee against the USD, and smuggled gold significantly affects India's jewellery export.

In conclusion, the study suggests that gold smuggling produces a tax loss for the government; however, this is highly negligible. At the same time, gold smuggling positively affects India's GDP, India's jewellery export, and the gold demand and supply. Smugglers try to take the advantage of arbitrage pricing to make more profit (Kumar 2017). The government should pay attention to the price spread between domestic and international prices. If the price spread is less, then there is less chance of making money by taking advantage of arbitrage pricing through smuggling. The price spread may be caused by the customs duty and the exchange rate of the Indian rupee. The customs duty can be minimized to encourage more gold imports through the formal process. This loss can be compensated when the gold is re-exported as jewellery to other countries, as there is a greater demand for Indian jewellery in other countries. This study is limited to the data

available on the Directorate of Revenue Intelligence website. There were difficulties with regard to the availability of sensitive data like the origin of gold smuggling, details of travellers, etc. These data could help the research into the use of advanced analytics models for a better prediction. In future, research can be undertaken to identify the problems faced by the Indian jewellers due to the smuggled gold, and how the smuggled gold is converted and enters into the official process of the business. This would be helpful for the bullion industry as well as for the policy makers.

**Author Contributions:** Conceptualization, M.I.S. and L.D.; methodology, L.D.; software, M.I.S.; validation, L.D.; formal analysis, M.I.S.; investigation, L.D.; resources, M.I.S. and L.D.; data curation, M.I.S.; writing—original draft preparation, M.I.S.; writing—review and editing, L.D.; visualization, M.I.S. and L.D.; supervision, L.D. All authors have read and agreed to the published version of the manuscript.

**Funding:** This research received no external funding.

**Data Availability Statement:** The data are sourced and compiled from the Directorate of Revenue Intelligence (DRI) website: https://dri.nic.in/, https://www.gold.org/; https://rbi.org.in/, (accessed on 30 March 2022).

**Acknowledgments:** We sincerely thank India Gold Policy Centre, Indian Institute of Management, Ahmedabad for conferring this paper as the "MCX-IIMA Award for Excellence in Research on Gold" at the 5th IGPC-IIMA Annual Gold & Gold Markets Conference 2022, held on 11–12 April 2022. We also thank Binu Zachariah, Department of English, Pondicherry University for the English revision.

**Conflicts of Interest:** The authors declare no conflicts of interest.

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
