# Peer review of "Gold Smuggling in India and Its Effect on the Bullion Industry"

_jrfm, doi:10.3390/jrfm17030122_

Round 1

Reviewer 1 Report

Comments and Suggestions for Authors

The authors did not actually identify the research gap, did not explain why the topic is important and whom the results of the research should be interesting for. The applied methods are rather simple and thus the results respresent generally a mere description of collected data. The article has a very strong potential, but it has not been exploited fully which is a huge opportunity missed by the authors.

Comments on the Quality of English Language

There are some major mistakes (missing words in sentences, frequent grammar issues, etc.) in the text which indicates the whole paper requires extensive editing.

Author Response

Comments from Reviewer 1

Comments and Suggestions for Authors

Comment 1

The authors did not actually identify the research gap, did not explain why the topic is important and whom the results of the research should be interesting for. The applied methods are rather simple and thus the results respresent generally a mere description of collected data. The article has a very strong potential, but it has not been exploited fully which is a huge opportunity missed by the authors.

Author’s Response: We sincerely appreciate the efforts taken by the reviewer for bringing out these points. We found studies on flow of illicit gold, country from which these gold are coming from and how India became the hub for gold smuggling but not a study found to establish the link between gold smuggling and its effect on bullion market. 

Based on the suggestions, we have now edited line 160 – 172.

We agree with the reviewer that the methods used are simple. It is being done deliberately to understand the pattern and characteristics as they are without going for advanced tools. However, we assure you to take up studies in future with advanced tools and compare the smuggling of gold in other countries as suggested by the reviewer.  

Indian Jewellers face many issues due to the smuggled gold. Hence, this study is an attempt to find out when and where the gold smuggling takes place in the country and how it is affected by the other variables with the simple statistical tools.

Comment 1

Comments on the Quality of English Language

There are some major mistakes (missing words in sentences, frequent grammar issues, etc.) in the text which indicates the whole paper requires extensive editing.

Author’s Response: We thank the reviewer for pointing out the errors. The manuscript was given to a Professor of English for proof reading and all the language errors have now been rectified.

Reviewer 2 Report

Comments and Suggestions for Authors

Dear authors,

Very interesting and useful topic of research. I believe it is an important topic of research and similar research could be done for other areas in the world. That being said I believe the manuscript should see some changes before publication. 

Introduction looks incomplete. It demonstrates a lot of information and data to introduce the topic however we have not been introduced to the gap in the literature. We also need a clear presentation of the aim of this research and a couple of statements to highlight the importance of the outcomes for the industry

Major Research Works Reviewed on Gold smuggling: Should be renamed as literature review or theoretical background: Much better section. It has a good flow and easy to follow. However, it is still lacking on information. Considering that this paper is based on secondary data literature review a table including previous research on the topic should be presented  in the literature review for visual representation. Abstract states that paper used several variables (quantity smuggled gold, exchange rates, major stock indices in the world, auspicious days in a month, domestic and international gold prices, India’s jewellery export, GDP, customs duty, and domestic gold supply), but I am not sure that these topics are mentioned or defined in the literature review. Finally, I believe we should have a visual model in the end of the literature review.

Data, Variables and Methodology: 3.3 Major Research Question section is not clear. Previous section (3.2 Objectives of the Study) presents the objectives of the study. I am not sure what is the purpose of 3.3 section, since it is mostly confusing the research purpose of this paper. I would recommend this section to be removed.

I believe this section is missing a paragraph to present the reliability (É‘) and the validity (AVE and/or CR) of the variables.  

Finally, “to suggest policy measures to control the grey market operation in India” is not really an aim but an outcome.

Results and Discussion: I believe it is a good section with a lot of results shown and some good discussion. The section would be even better if some of these topics were better discussed in the literature review and compared with the results in this section.

Conclusion: We have an aim here, so I recommend this to be mentioned in the introduction from the beginning. I recommend this section to include limitations, future research and practical and theoretical contributions. 

Comments on the Quality of English Language

I believe that this paper should be proofread for the quality of English language and citations. For example:

I would like to ask words such as “our” (see line 31) to be removed

Line 116 “as stated in the introduction” should be removed

Author Response

Comments from Reviewer 2:

Comments and Suggestions for Authors

Comment 1.

Very interesting and useful topic of research. I believe it is an important topic of research and similar research could be done for other areas in the world. That being said I believe the manuscript should see some changes before publication. 

Introduction looks incomplete. It demonstrates a lot of information and data to introduce the topic however we have not been introduced to the gap in the literature. We also need a clear presentation of the aim of this research and a couple of statements to highlight the importance of the outcomes for the industry

Author’s Response: We sincerely appreciate the efforts taken by the reviewer for bringing out these points. Review of literature show that studies on flow of illicit gold, country from which these golds are coming from and how India became a hub for gold smuggling, but not a study found to establish the link between gold smuggling and its effect on bullion market.  Therefore, we were not able to highlight it earlier, however now we incorporated. The purpose of this research basically to understand the pattern and characteristics of Gold Smuggling in India and its effect on the Bullion Market.

Based on the suggestions, we have now edited line 160 – 173.

Comment 2.

Major Research Works Reviewed on Gold smuggling: Should be renamed as literature review or theoretical background: Much better section. It has a good flow and easy to follow. However, it is still lacking on information. Considering that this paper is based on secondary data literature review a table including previous research on the topic should be presented in the literature review for visual representation. Abstract states that paper used several variables (quantity smuggled gold, exchange rates, major stock indices in the world, auspicious days in a month, domestic and international gold prices, India’s jewellery export, GDP, customs duty, and domestic gold supply), but I am not sure that these topics are mentioned or defined in the literature review. Finally, I believe we should have a visual model in the end of the literature review.

Author’s Response: Thank you for the suggestion. We have changed the title 2 to Literature Review. The variables used are of relevant to the industry and suggestions given by the Indian Gold Policy Center which has experts to guide the research. Based on their suggestions all these variables are included not based on literature. Due to page constraints we are not able to go for visual model of literature.

Data, Variables and Methodology: 3.3 Major Research Question section is not clear. Previous section (3.2 Objectives of the Study) presents the objectives of the study. I am not sure what is the purpose of 3.3 section, since it is mostly confusing the research purpose of this paper. I would recommend this section to be removed.

Author’s Response: Thank you for pointing out this. We agree with the reviewer. As suggested by the reviewer, we have now removed the section 3.3 Major Research Questions.

I believe this section is missing a paragraph to present the reliability (É‘) and the validity (AVE and/or CR) of the variables.  

Author’s Response: We agree with the reviewer. We have used time series data, hence all the variables are tested for its stationarity. All the variables follow the same order of integration.

Finally, “to suggest policy measures to control the grey market operation in India” is not really an aim but an outcome.

Author’s Response: Thank you for the suggestion. We have made the changes accordingly in the manuscript (Line no 211)

Results and Discussion: I believe it is a good section with a lot of results shown and some good discussion. The section would be even better if some of these topics were better discussed in the literature review and compared with the results in this section.

Author’s Response: Thank you for the comment. But existing literature did not talk about the variables that we included in the analysis. It is an attempt what we have made. We completely relied upon the opinion of expert’s from India Gold Policy Centre for including and excluding the variables. Therefore, results are presented are not compared with the existing literature. Wherever results are in conformance with other studies, they are being cited in the manuscript.

Conclusion: We have an aim here, so I recommend this to be mentioned in the introduction from the beginning. I recommend this section to include limitations, future research and practical and theoretical contributions. 

Author’s Response: We appreciate the reviewer’s suggestion. We have now added in line 463 - 469  

Comments on the Quality of English Language

I believe that this paper should be proofread for the quality of English language and citations. For example:

Author’s Response: We thank the reviewer for pointing out the errors. The manuscript was given to a Professor of English for proof reading and all the language errors have now been rectified.

I would like to ask words such as “our” (see line 31) to be removed

Author’s Response: Thank you. This paper is prepared for the Indian bullion industry. Hence, we used this word in the manuscript. We have now removed this word “our” found throughout in the manuscript in the following lines:

Line 31

Line 116 “as stated in the introduction” should be removed

Author’s Response: Thank you. We have removed this. Revised line (117).

Reviewer 3 Report

Comments and Suggestions for Authors

I appreciate the effort involved for this study named “Gold Smuggling in India and its Effect on Bullion Industry” and congratulate the authors for this, but from my point of view the manuscript is in danger by self-plagiarism (more than 80%). In order not to deceive (you, reviewers and readers) properly quoted, cited and correct acknowledgment shell be use in the manuscript. Some materials can be reused, but you should cited the old article because for the present article have less coauthors (is missing one author from the first material, Rakshambiga VN II PGDM Student , St. Joseph’s Institute of Management. I suggest updating the material and the resources as well as the references because there is just one paper from the current year 2023 added to the old reference from the first article with the same title which was primary sustained in 12.04.2022 at the 5th Annual Gold and Gold Markets Conference and publish on the webpage of Indian Institute of Management Ahmedabad. 

Comparations between the two articles show: 60% of the material is identical, 12,5% present minor changes, 6,5% is paraphrased.

Author Response

Comments from Reviewer 3:

Comments and Suggestions for Authors

I appreciate the effort involved for this study named “Gold Smuggling in India and its Effect on Bullion Industry” and congratulate the authors for this, but from my point of view the manuscript is in danger by self-plagiarism (more than 80%). In order not to deceive (you, reviewers and readers) properly quoted, cited and correct acknowledgment shell be use in the manuscript. Some materials can be reused, but you should cited the old article because for the present article have less coauthors (is missing one author from the first material, Rakshambiga VN II PGDM Student , St. Joseph’s Institute of Management. I suggest updating the material and the resources as well as the references because there is just one paper from the current year 2023 added to the old reference from the first article with the same title which was primary sustained in 12.04.2022 at the 5th Annual Gold and Gold Markets Conference and publish on the webpage of Indian Institute of Management Ahmedabad. 

Comparations between the two articles show: 60% of the material is identical, 12,5% present minor changes, 6,5% is paraphrased.

Author’s Response: We sincerely thank the reviewer for spending his time in reviewing our manuscript and bringing out this comment.

We agree with the reviewer’s comments. The paper reviewed was presented at the 5th Annual Gold and Gold Markets Conference in the year 2022 with three authors. Based on the comments received from the conference, it is fine tuned by two authors as the third author has left after completing the course.  Moreover, the third author did not have any role in completing this paper for publication. The conference organizers have published the paper which was presented in the conference as a working paper with the condition that authors can publish it anywhere else. Precisely because of this 80% plagiarism would have been there. However, we modified lot more contents in this paper. We like to inform you that this paper has been adjudged as BEST PAPER and given an award.

As per the instructions received from the Managing editor, We will be submitting the authorship form to avoid future misunderstanding.

Round 2

Reviewer 3 Report

Comments and Suggestions for Authors

It is appreciated that you admit this article it is based (for a categorically significant percent) on a previous conference paper created by 3 co-authors and published online by Indian Institutes of Management Ahmedabad https://www.iima.ac.in/sites/default/files/2023-06/Maria%20Immanuvel.pdf 

with these three names as authors, and then as you reply, “it is fine-tuned by two authors”.

It could be unfair (for the readers and your co-author Rakshambiga VN) not even to mention somehow the "working" papers that you based this article on.

Good fortune!

Author Response

Author’s Response: We sincerely appreciate the reviewer for checking the plagiarism of the manuscript. As a researcher, we completely agree with the reviewer’s comments and it would be unfair if we publish the manuscript without her knowledge. We understand that it affects the integrity of the journal also.  

Hence, as per the instructions received from the editor, we have submitted the authorship forms. Also, the third author Rakshambiga VN has agreed and given her consent for publishing the paper without her name. The screen shots of No Objection e mail received from Rakshambiga VN is given in the attached file.
